# Immunogenicity of BNT162b2 mRNA-Based Vaccine against SARS-CoV-2 in People with Cystic Fibrosis According to Disease Characteristics and Maintenance Therapies

**DOI:** 10.3390/biomedicines10081998

**Published:** 2022-08-17

**Authors:** Gianfranco Alicandro, Valeria Daccò, Lisa Cariani, Chiara Rosazza, Calogero Sathya Sciarrabba, Federica Ferraro, Chiara Lanfranchi, Paola Medino, Daniela Girelli, Carla Colombo

**Affiliations:** 1Department of Pathophysiology and Transplantation, Università degli Studi di Milano, 20122 Milan, Italy; 2Cystic Fibrosis Centre, Fondazione IRCCS Ca’ Granda Ospedale Maggiore Policlinico, 20122 Milan, Italy; 3Microbiology Unit, Fondazione IRCCS Ca’ Granda Ospedale Maggiore Policlinico, 20122 Milan, Italy

**Keywords:** cystic fibrosis, SARS-CoV-2, COVID-19, vaccine, antibody response, humoral response

## Abstract

During the SARS-CoV-2 vaccination campaign, people with CF (pwCF) were considered a clinically vulnerable population. However, data on the immunogenicity of anti-SARS-CoV-2 vaccines in pwCF are lacking. We conducted a prospective study enrolling all patients aged > 12 and who were followed-up in our CF center and received two doses of the BNT162b2 vaccine in the period of March–October 2021. Blood samples were taken from them for the quantification of antibodies to the SARS-CoV-2 spike protein receptor binding domain immediately before receiving the first dose and 3 and 6 months after the second dose. We enrolled 143 patients (median age: 21 years, range: 13–38), 16 of whom had had a previous infection. Geometric mean antibody titer (GMT) 3 months after vaccination was 1355 U/mL (95% CI: 1165–1575) and decreased to 954 U/mL (95% CI: 819–1111) after 6 months (*p* < 0.0001). GMT was higher among previously infected patients as compared to those naïve to SARS-CoV-2 (6707 vs. 1119 U/mL at 3 months and 4299 vs. 796 U/mL at 6 months, *p* < 0.0001) with no significant differences in the rate of decline over time (*p* = 0.135). All pwCF mounted an antibody response after two doses of the BNT162b2 vaccine, which waned at 6 months from vaccination. Age ≥ 30 years and the use of inhaled corticosteroids were associated with a lower humoral response. Between the second and the third doses, nine episodes of vaccine breakthrough infections were observed.

## 1. Introduction

Cystic fibrosis (CF) is the most frequent life-threatening genetic disease among Caucasians and is caused by mutations in the CF Transmembrane Conductance Regulator (CFTR) gene. The gene codes the CFTR protein, an ion channel that regulates chloride and bicarbonate traffic at the cell surface, and its abnormal function causes the production of thick secretions in many organs, including the pancreas and the lungs. Clinical manifestations of CF mainly include fat malabsorption, which requires pancreatic enzyme replacement therapy, and frequent respiratory infections that lead to lung damage and the eventual progression to end-stage lung disease and the need for lung transplantation [1].

People with CF (pwCF) are considered a clinically vulnerable population, which is why they were vaccinated at the very beginning of the vaccination campaign in many countries, including Italy (February–March 2021) [2]. However, data on how this population responded to the vaccination are limited to a small study on 33 subjects, who showed higher antibody responses as compared to a group of healthy controls [3].

Low seroconversion rates were reported in pwCF during the 2009/H1N1 pandemic, especially in transplanted patients and in those with a low body mass index (BMI) [4], which suggests a potential defect in the immune response to the influenza vaccine and may be linked to the defective expression of CFTR in lymphocytes [5,6]. In addition, innate and adaptive immunity was dysregulated in this population due to inherited and acquired factors, including the epithelial barrier function, pathogen sensing, leukocyte and phagocyte recruitment, and communication between innate and adaptive immunity [7,8]. Moreover, the maintenance therapy of pwCF may involve the administration of several inhaled and systemic drugs that may affect the immunological response to vaccines. Among these drugs, steroids are frequently prescribed to reduce lung inflammation and to treat wheezing; pwCF also receive frequent antibiotic courses to treat respiratory exacerbations and to eradicate pathogenic bacteria from the respiratory airways. Antibiotics affect the gut microbiome, and microbiome alteration has been linked to the reduced immunogenicity and efficacy of vaccines [9,10]. In a controlled experiment [10], broad-spectrum antibiotics (vancomycin, neomycin, and metronidazole) administered to healthy young adults before and after seasonal influenza vaccination resulted in a transient decrease in gut bacterial load and in a long-lasting reduction in microbiome diversity. These changes were mirrored by reduced antibody responses in individuals with low pre-existing immunity to the influenza virus.

This study aims to evaluate the antibody response to the BNT162b2 mRNA-based vaccine against SARS-CoV-2 in pwCF and to characterize subgroups of this population with low responses.

## 2. Materials and Methods

All patients aged > 12 years who were in regular follow-up at the Reference Centre for CF of the Lombardia region and had received two doses of the mRNA-based vaccine BNT162b2 (COMIRNATY BioNTech Manufacturing GmbH, Mainz, Germany) between March and October 2021 were included in this prospective study. They underwent 3 blood sample tests for the quantification of antibodies to the SARS-CoV-2 spike protein receptor binding domain (S1-RBD). Serum titers were measured immediately before administering the first dose of the vaccine and at 3 and 6 months after receiving the second dose using an electrochemiluminescence immunoassay (Elecsys Anti-SARS-CoV-2 S Roche Diagnostics, Monza, Italy) (positive cutoff: 0.8, lower limit of quantification; LLOQ: 0.4 U/mL, upper limit of quantification; ULOQ: 12,500 U/mL; sensitivity: 98.8%, and specificity: 100%). Values below the LLOQ were set to LLOQ/2, and values above the ULOQ were set to ULOQ before analysis. This serological assay used a recombinant protein representing the S-RBD protein in a one-step double antigen sandwich assay format. Samples were incubated with a mix of biotinylated and ruthenylated RBD antigen, and double-antigen sandwich immune complexes were formed when corresponding antibodies were present. After the addition of streptavidin-coated microparticles, the DAGS complexes bound to the solid phase via the interaction of biotin and streptavidin. The microparticles were magnetically captured on the surface of the electrode; electrochemiluminescence was induced by applying a voltage and was measured with a photomultiplier. Samples with a value ≥ 0.8 U/mL were considered “reactive” (positive). The analyses were performed at the Clinical Laboratory of Fondazione IRCCS Ca’ Granda Ospedale Maggiore Policlinico, Milan, Italy.

Information on prior infection by SARS-CoV-2, including via positive reverse transcriptase-polymerase chain reaction (RT-PCR) molecular test and the occurrence of COVID-19 symptoms, was collected through a telephone interview carried out by the attending physicians, while clinical data and microbiological results of sputum cultures were retrieved from patient medical records. During the interview, a diary card recording the solicited local and systemic adverse reactions were also completed for 7 days after each administration. Severity of adverse reactions was graded using the following criteria: mild (transient or mild discomfort for <48 h, no interference with activity, and no medical intervention or therapy required), moderate (mild-to-moderate limitation in activity, and no or minimal medical intervention or therapy required), severe (substantial limitation in activity and medical intervention or therapy required), or potentially life-threatening (requiring assessment in emergency department or admission to hospital) [3].

In patients naïve to SARS-CoV-2, antibody titers were compared across groups defined by demographic characteristics (sex and age), indicators of disease severity (pancreatic insufficiency, underweight, infection by *P*. *aeruginosa*), and current CF maintenance therapies, including inhaled and systemic antibiotics, azithromycin, inhaled corticosteroids (ICS), and CFTR modulators. Humoral response was also compared among patients reporting moderate/severe adverse reactions vs. those who had none or only mild reactions and among patients with systemic vs. local reactions after the administration of the vaccine.

Exocrine pancreatic status was determined according to fecal elastase-1 levels, with levels < 200 µg/g being indicative of pancreatic insufficiency. Patients were considered underweight if their BMI-for-age z-score was <−1.64 (i.e., <5th percentile) (for patients aged ≤ 20 years), or if their BMI was <18.5 kg/m^2^ (for older patients) [11]. Z-scores of BMI were obtained using the Italian reference data [12]. *P*. *aeruginosa* infection was defined by a positive sputum culture during the last visit preceding vaccination.

Given the skewed distribution, antibody titers were summarized using geometric means and 95% confidence intervals (CI). Antibody titers at 3- and 6-month post-vaccination and their variations were compared across patient groups (defined by demographic and clinical characteristics, maintenance therapies, and occurrence of adverse reactions to vaccination) using linear mixed-effect models with subject random intercept. The model included the log10-transformed antibody titer as a dependent variable and the group variable, the time (3 or 6 months), and their interaction as fixed effects. The statistical significance of the fixed effects was evaluated using the likelihood ratio test.

To test the independent associations between the explanatory variables and antibody response, we fitted a multivariable linear mixed-effect model, including all the factors significantly associated with antibody response in the analysis described above. The estimated β coefficients and 95% CI were used to evaluate the effect of each factor on the log10-transformed antibody titer.

Cumulative incidence of post-vaccination SARS-CoV-2 infection was computed using the complement of the Kaplan–Meier estimator, with the time elapsed since the second dose as the time scale. Post-vaccination infection was defined as a reported positive result from a nasopharyngeal swab RT-PCR for SARS-CoV-2 or from an antigen test between the second and the third dose. Patients were censored at the date of the third dose. A Cox regression model was used to estimate the hazard ratio (HR) of infection associated with the log10-transformed antibody titer measured 6 months after the second dose of the vaccine.

All tests were two-sided, with the significance level set at 0.05.

## 3. Results

Table 1 summarizes the main demographic and clinical characteristics of the 143 pwCF who underwent serological tests immediately before vaccination and 3 and 6 months after the second dose of the vaccine, as well as the type and severity of adverse reactions. The study population included mainly adults (86%) with a mild to moderate disease severity, as indicated by the relatively low percentage of patients with pancreatic insufficiency (55.9%) and the relatively high values of ppFEV1, with most patients having predicted values ≥ 80%. Almost 40% had a respiratory infection by *P. aeruginosa*, 45.5% were taking inhaled steroids, 35.7% azithromycin, 27.3% other systemic antibiotics, and 25.2% were being treated with CFTR modulators. Only a minority was taking systemic steroids (n = 3, 2.1%). Most patients (95.1%) reported either local or systemic reactions after the first or the second dose; however, these were of mild severity in half of the patient population and of moderate severity in around 40%. None had severe adverse reactions (Table 1).

Figure 1 shows the distributions of antibody titers prior to vaccination and 3 and 6 months after the second injection in patients naïve to SARS-CoV-2 as well as in those with previous exposure to the virus. All patients seroconverted after vaccination (anti-S-RBD ≥ 0.8 U/mL), with antibody titers surging at 3 months from vaccination in both groups and reaching higher values in people with past exposure to SARS-CoV-2 as compared to patients naïve to the virus. At 3 months, the mean antibody titer was 1117 U/mL (95% CI: 983–1271) in patients naïve to the virus and 6707 U/mL (95% CI: 4666–9641) in those who had been exposed to the virus. At 6 months, the antibody titers significantly decreased in both groups, with mean values of 796 U/mL (95% CI: 695–911) in patients naïve to the virus and 4299 U/mL (2943–6280) in those previously infected by SARS-CoV-2.

Table 2 shows the antibody titer at 3- and 6-month post-vaccination according to selected demographic and clinical characteristics as well as the CF maintenance therapies in people naïve to SARS-CoV-2. There was no difference between sexes, while antibody response decreased with increasing age, although the waning rate was similar across age groups. Patients with pancreatic insufficiency showed a lower antibody response as compared to those with pancreatic sufficiency, although a more rapid decline at 6 months was observed among patients with pancreatic sufficiency. Antibody titers were also lower in patients with *P*. *aeruginosa* infection than in those who were *P*. *aeruginosa*-free. The antibody response of underweight patients was comparable to that observed in patients with normal nutritional status.

There was no significant difference among patients who were taking inhaled antibiotics or azithromycin and those who were not, while humoral response was lower in patients treated with other systemic antibiotics as well as in those taking inhaled corticosteroids or receiving CFTR modulators. The relationship between oral corticosteroids and antibody response could not be evaluated since only three patients were taking oral corticosteroids; their antibody titers were: 453, 666 and 1883 U/mL at 3 months after vaccination and 258, 533 and 1473 at 6 months after vaccination.

Table 3 gives the results of the model, including all the potential determinants of antibody response. Past infection by SARS-CoV-2, age ≥ 30 years, time of the measurements, and treatment with inhaled corticosteroids were the major independent determinants of the antibody response to two doses of the BNT162b2 vaccine in pwCF.

Figure 2 shows the individual antibody titer measured at 3- and 6-month postvaccination and the predicted responses according to age group and treatment with inhaled corticosteroids.

Type of adverse reactions (systemic vs. local/none) and their severity did not significantly affect antibody titers (Table 4).

Figure 3 shows the cumulative incidence of post-vaccination SARS-CoV-2 infection after the second dose of the vaccine. Among the 142 patients with available information on infection status at follow-up, 9 patients were infected by SARS-CoV-2 before receiving the third dose of the vaccine. The cumulative incidence at day 240 was 1.6% (95% CI: 0–3.8%) and at the last observed event time (day 264) was 15.6% (95% CI: 4.8–25.1%). However, by that time, most patients had received the third dose. The majority of infections (n = 7, 77.8%) occurred between day 240 and day 264, 2 infections occurred between day 180 and 240, none before day 180. Eight patients had mild COVID-19, and one required hospitalization. Antibody titer at 6 months from the second dose was not associated with a significant reduction in the infection rate (HR for one-point increment in log-10 antibody titer: 0.91, 95% CI: 0.20–4.21), although this analysis is limited by the low number of events.

## 4. Discussion

In this relatively large population of pwCF, the administration of two doses of the mRNA-based vaccine BNT162b2 induced a strong antibody response, especially among patients previously infected, which waned within 6 months from the second dose of the vaccine. The antibody titer was highly variable, and a few subgroups of the population had lower responses, including patients aged ≥ 30 years, those with pancreatic insufficiency and *P*. *aeruginosa* infection, and patients regularly treated with systemic antibiotic, inhaled corticosteroids, or CFTR modulators. When we considered all these factors together, we found that previous infection, age, and treatment with inhaled corticosteroids were the only independent predictors of the humoral response to two injections of the BNT162b2 vaccine. This indicates that the associations with the remaining factors were largely mediated by age and disease severity.

The antibody titers and their dynamics over 6 months post-vaccination that were detected in our pwCF were comparable, on average, to what has been observed in non-CF populations [13,14]; on the other hand, the lower response observed in subjects aged 30–38 years is peculiar to CF. This finding may be attributed, at least in part, to the progressive nature of CF, characterized by chronic inflammation, frequent respiratory infections, lung damage, and prolonged treatments with antibiotics. All these factors may affect the ability of the immune system to mount a humoral response to the vaccine, especially in older patients, who are more likely to have a more severe expression of the disease.

The higher response detected among patients previously infected by SARS-CoV-2 is comparable to what has been reported in non-CF subjects after the first dose of mRNA-based vaccines that led to the recommendation of a single dose in pre-exposed healthy individuals [15,16].

The use of corticosteroids over a prolonged period of time is associated with well-known, significant side effects. Patients treated with systemic corticosteroids mount a lower humoral response to vaccination, including that against SARS-CoV-2. Studies on patients with musculoskeletal diseases, cancer, and on transplant recipients documented that long-term treatment with oral corticosteroids induce a lower antibody response to mRNA-based vaccines [17,18,19]. However, whether the prolonged use of ICS has any immunosuppressive properties, and specifically if ICS affects the humoral response to vaccines, is unclear, especially in pwCF. Using models of rhinovirus infection, Singanayagam et al. found that the ICS fluticasone propionate impairs the innate and acquired antiviral immune responses during virus-induced chronic obstructive pulmonary disease (COPD) exacerbations [20]. However, controversial evidence has emerged in studies on asthmatic patients, where prolonged (≥6 months) treatment with ICS had no effect on cellular immunity [21,22]. Moreover, a study on children and adults with asthma found that the use of low dose ICS (≤504 µg/day in adults or 336 µg/day in children of beclomethasone dipropionate equivalent) did not adversely affect the humoral response to influenza A (H1N1, H3N3) vaccine antigens [23]. Similar results were found in a study based on elderly patients with COPD, where patients who received daily treatment with ICS (any dose of beclomethasone, budesonide, or fluticasone) showed antibody responses to a MF59-ajuvanted vaccine against influenza strains (A/H1N1, A/H3N, and B) comparable to patients who did not receive any steroid treatment [24]. These findings are likely due to the smaller dose adsorbed in patients taking ICS as compared to those treated with oral therapy. However, further investigation is needed to clarify the potential role of ICS in the humoral response to vaccines.

Contrary to what has been documented in a study based on 578 healthcare workers [15], which found a higher antibody response at 12–19 days after BNT162b2 or mRNA-1273 vaccination, we did not observe any relationship between the occurrence of systemic reactions and antibody titer. However, the lack of association between symptom severity and vaccine-induced antibody response was previously reported in a study based on 206 healthy adults with no history of COVID-19 [25].

A potential link between malnutrition and vaccine responsiveness in pwCF has been suggested by Launay et al., who found a lower immune response to the vaccine among pwCF with low BMI during the 2009 A/H1N1 pandemic [4]. We could not confirm this finding; however, there were only eight underweight patients in our cohort.

Our data suggest that the vaccine provided adequate protection against SARS-CoV-2 infection in our patient population during the 6 months following the second injection. However, they also confirm that the protection is only transient since vaccine breakthrough infections occurred after that period.

When interpreting our results, it should be noted that the decreased antibody titers 6 months after vaccination does not necessarily reflect reduced protection against infection and severe COVID-19. In fact, T cell response as well as antiviral B and T cell memory, which were not measured in our study, have proven to be important in the maintenance of SARS-CoV-2 immunity despite the drop in circulating antibodies [26,27,28,29].

Our study provides unique data on the durability and dynamics of humoral responses to the most frequently administered mRNA-based vaccine among pwCF. These data are of clinical relevance since the levels of antibodies binding the S-RBD antigen measured post-vaccination are related to a lower probability of infection [30].

However, our population of pwCF is characterized by mild to moderate disease, and we could not evaluate the antibody response in patients with severe CF, including those with end-stage lung disease and transplanted patients, with the latter expected to have a lower humoral response due to strong immunosuppressive therapies. Other limitations of our study include the lack of a control group and the short-term evaluation of the durability of the antibody response (6 months). Finally, our data were collected during the pre-Omicron phase of the pandemic, when two doses of the vaccine had been administered. At the time of writing, all patients had received the third dose, the immunogenicity of which in this population remains to be determined.

## 5. Conclusions

The immunogenicity of the BNT162b2 vaccine in pwCF was comparable to that observed in the general population. However, we found a marked heterogeneity in our patients, with lower humoral responses in patients aged ≥ 30 years and those using ICS. Future studies focusing on mechanisms other than antibody production are needed to further characterize the durability of immunization in pwCF, also in view of the new emerging variants that seem to partly escape from antibody neutralization [31,32,33].

## Figures and Tables

**Figure 1 biomedicines-10-01998-f001:**
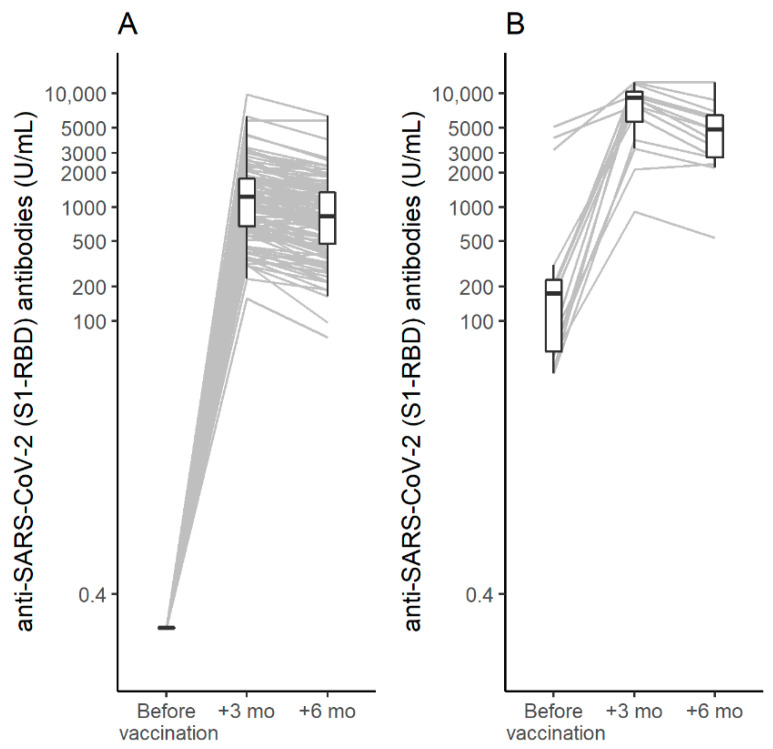
Antibody titer before vaccination and 3 and 6 months after the second dose of the BNT162b2 vaccine: (**A**) in people with cystic fibrosis naïve to the virus; (**B**) in people with cystic fibrosis previously infected by SARS-CoV-2. Grey lines indicate individual antibody responses, while box plots show their distribution at different time points (before vaccination, 3 and 6 months after second injection).

**Figure 2 biomedicines-10-01998-f002:**
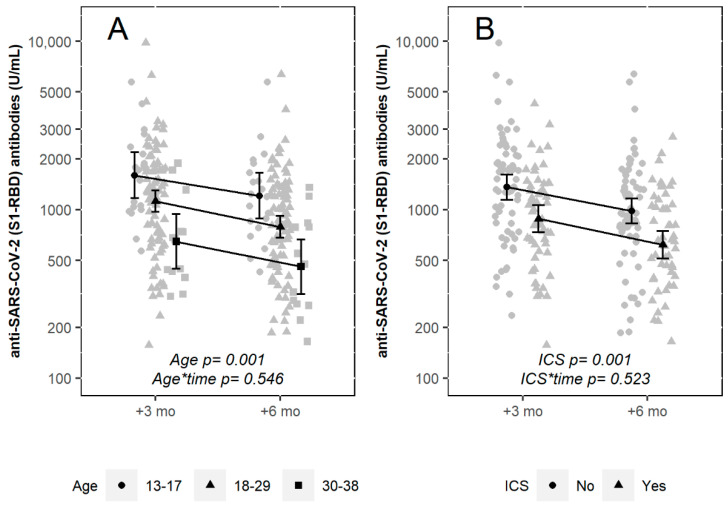
Antibody titer at 3 and 6 months after the second dose of the BNT162b2 vaccine in people with cystic fibrosis naïve to SARS-CoV-2, according to age group (**A**) and inhaled corticosteroid treatment (**B**). Grey symbols indicate individual data, black symbols and error bars show the estimated mean values and corresponding 95% confidence intervals obtained from mixed-effect regression models. *p*-values indicate the statistical significance of the main effects of the age group and inhaled corticosteroid treatment and their interaction with time (3- or 6-month post-vaccination).

**Figure 3 biomedicines-10-01998-f003:**
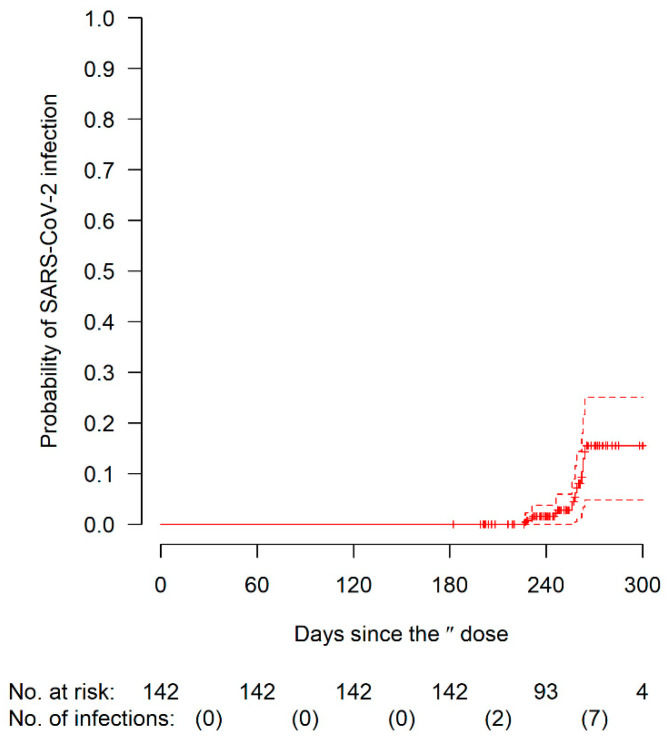
Cumulative incidence of SARS-CoV-2 infection occurred in the interval between the second and the third dose of the BNT162b2 vaccine in people with cystic fibrosis. Dashed lines are 95% confidence intervals of the cumulative incidence function (continuous line). The numbers below the figure are: number at risk at the beginning of the interval and number of infections during the interval. Follow-up infection status was not available for one patient.

**Table 1 biomedicines-10-01998-t001:** Characteristics of the study population.

Number of patients	143 (100)
Male sex	75 (52.4)
Age	
Median (IQR)	21 (18–25)
Adults	123 (86.0)
Age group: 13–17 years	20 (14.0)
Age group: 18–29 years	107 (74.8)
Age group: 30–38 years	16 (11.2)
Pancreatic insufficiency	80 (55.9)
*P. aeruginosa* infection	57 (39.9)
BMI, kg/m^2^, median (IQR) ^a^	22.4 (20.1; 24.4)
BMI, z-score, median (IQR) ^a,b^	0.10 (−0.55; 0.81))
Underweight ^a,c^	8 (5.6)
ppFEV1 ^d^	
Median (IQR)	97 (82–106)
≥80%	107 (75.9)
40–79%	33 (23.4)
<40%	1 (0.7)
Maintenance therapies	
Inhaled antibiotics	24 (16.8)
Systemic antibiotics	39 (27.3)
Azithromycin	51 (35.7)
Inhaled corticosteroids	65 (45.5)
Systemic corticosteroids	3 (2.1)
CFTR modulators ^e^	36 (25.2)
Oxygen therapy	1 (0.7)
Prior SARS-CoV-2 infection	
Yes	16
RT-PCR confirmed infection	1
Symptomatic infection	1
Unknown	4
Adverse reactions after the first or second dose of the BNT162b2 vaccine	
None	7 (4.9)
Local	128 (89.5)
Systemic	103 (72.0)
Mild	71 (49.7)
Moderate	57 (39.9)
Severe	0

BMI: Body Mass Index. IQR: Interquartile Range. RT-PCR: Real-Time Reverse Transcription–Polymerase Chain Reaction. SARS-CoV-2: Severe Acute Respiratory Syndrome Coronavirus 2. Data are expressed as numbers (%), unless otherwise indicated. ^a^ BMI was not available for one patient due to missing value for height. ^b^ For patients aged > 20 years, the BMI z-score was obtained using the reference values corresponding to the age of 20 years. ^c^ Underweight was defined according to sex and an age-specific BMI z-score < −1.64 (i.e., <5th percentile) for patients aged ≤ 20 years and a BMI < 18.5 kg/m^2^ for older patients. ^d^ Not available for two patients. ^e^ A total of 31 patients were treated with lumacaftor + ivacaftor, 1 patient with tezacaftor + ivacaftor, and 4 patients with elexacaftor + tezacaftor + ivacaftor.

**Table 2 biomedicines-10-01998-t002:** Antibody titer at 3 and 6 months after the second dose of the BNT162b2 vaccine in people with cystic fibrosis naïve to SARS-CoV-2, according to demographic and clinical characteristics and cystic fibrosis maintenance therapies.

Group	No.	3 Months from Second Dose	6 Months from Second Dose	*p*-Value for the Main Effect ^a^	*p*-Value for the Interaction with Time ^a^
Sex				0.494	0.694
Males	62	1174 (973–1417)	829 (680–1010)		
Females	61	1063 (892–1267)	764 (634–921)		
Age group				0.001	0.546
13–17 years	20	1600 (1235–2074)	1209 (924–1581)		
17–29 years	89	1123 (965–1307)	790 (675–926)		
29–38 years	14	647 (477–878)	458 (325–647)		
Pancreatic insufficiency				<0.0001	0.035
No	51	1561 (1313–1857)	1049 (880–1250)		
Yes	72	882 (750–1036)	655 (545–786)		
Underweight ^b^				0.411	0.555
No	114	1095 (956–1253)	776 (673–896)		
Yes	8	1324 (900–1950)	994 (722–1370)		
*P. aeruginosa* infection				0.001	0.178
No	72	1354 (1153–1592)	940 (795–1111)		
Yes	51	852 (706–1028)	630 (509–778)		
Inhaled antibiotics				0.155	0.389
No	101	1175 (1018–1356)	829 (715–961)		
Yes	22	888 (676–1166)	661 (473–922)		
Systemic antibiotics				0.004	0.409
No	87	1272 (1096–1475)	895 (766–1045)		
Yes	36	818 (653–1023)	600 (467–772)		
Azithromycin				0.057	0.073
No	77	1239 (1049–1464)	854 (716–1018)		
Yes	46	939 (774–1140)	707 (574–872)		
Inhaled corticosteroids				0.001	0.538
No	67	1360 (1143–1618)	983 (825–1170)		
Yes	56	883 (743–1050)	619 (510–751)		
CFTR modulators				0.005	0.16
No	89	1266 (1088–1471)	883 (756–1033)		
Yes	34	807 (652–998)	606 (471–779)		

Data are geometric means (95% CI). ^a^
*p*-values were obtained from mixed-effect regression models, including the logarithm of the antibody titer as a response variable, the main effects of the group variable and time, and a group-by-time interaction. ^b^ Underweight was defined according to sex and an age-specific BMI z-score < −1.64 (i.e., <5th percentile) for patients aged ≤ 20 years and a BMI < 18.5 kg/m^2^ for older patients. BMI was not available for one patient.

**Table 3 biomedicines-10-01998-t003:** Determinants of antibody response to two doses of the BNT162b2 vaccine in people with cystic fibrosis.

Potential Determinant of Antibody Response	β Coefficients ^a^	95% CI	*p*-Value ^b^
Intercept	3.368	(3.231 to 3.505)	
Age group:18–29 vs. 13–17	−0.142	(−0.283 to −0.002)	0.047
Age group: 30–38 vs. 13–17	−0.354	(−0.552 to −0.157)	<0.001
PI vs. PS	−0.089	(−0.215 to 0.037)	0.165
Pa infection (Yes vs. No)	−0.081	(−0.200 to 0.037)	0.177
SAB (Yes vs. No)	−0.068	(−0.188 to 0.052)	0.267
ICS (Yes vs. No)	−0.121	(−0.225 to −0.018)	0.022
CFTRmod (Yes vs. No)	−0.046	(−0.169 to 0.078)	0.469
Prior infection by SARS-CoV-2 (Yes vs. No)	0.704	(0.552 to 0.857)	<0.001
Time from second injection: 6 vs. 3 months	−0.153	(−0.172 to −0.133)	<0.001

CI: Confidence Interval. CFTRmod: CFTR Modulators. ICS: Inhaled Corticosteroids. PI: Pancreatic Insufficiency. Pa: *Pseudomonas aeruginosa*. PS: Pancreatic Sufficiency. SAB: Systemic Antibiotics. ^a^ β coefficients indicate the expected difference in the mean log10-transformed antibody titer estimated by a mixed-effect regression model with a subject-specific random intercept. ^b^
*p*-value indicates whether beta is significantly different from 0 (Wald’s test).

**Table 4 biomedicines-10-01998-t004:** Antibody titer at 3 and 6 months after the second dose of the BNT162b2 vaccine in people with cystic fibrosis naïve to SARS-CoV-2, according to the severity and type of adverse reactions.

Group	No.	3 Months from Second Dose	6 Months from Second Dose	*p*-Value for the Main Effect ^a^	*p*-Value for the Interaction with Time ^a^
Severity of adverse reactions				0.526	0.576
None/Mild	71	1165 (974–1392)	820 (678–993)		
Moderate	52	1056 (880–1268)	764 (633–922)		
Systemic reactions				0.819	0.226
No	32	1119 (856–1463)	836 (623–1123)		
Yes	91	1117 (964–1293)	782 (672–911)		

^a^*p*-values were obtained from mixed-effect regression models, including the logarithm of the antibody titer, the main effects of the group variable and time, and a group-by-time interaction.

## Data Availability

The data presented in this study are available on request from the corresponding author. The data are not publicly available due to privacy and ethical restrictions.

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
