# Peer review of "Immunogenicity of BNT162b2 mRNA-Based Vaccine against SARS-CoV-2 in People with Cystic Fibrosis According to Disease Characteristics and Maintenance Therapies"

_biomedicines, 2022, doi:10.3390/biomedicines10081998_

Round 1

Reviewer 1 Report

Introduction

“A potential defect in the immune response to influenza vaccine was suggested dur- 39 ing the 2009/H1N1 pandemic [2], which may be linked to the defective expression of CFTR 40 in lymphocytes [3,4]. In addition, in this population, innate and adaptive immunity is 41 dysregulated due to inherited and acquired factors, including epithelial barrier function, 42 pathogen sensing, leukocyte and phagocyte recruitment and communication between in- 43 nate and adaptive immunity [5,6] Moreover, the maintenance therapy of pwCF may in- 44 volve the administration of several inhaled and systemic drugs that may affect the 45 Citation: Lastname, F.; Lastname, F.; Lastname, F. Title. Biomedicines 2022, 10, x. https://doi.org/10.3390/xxxxx Academic Editor: Firstname Last[1]name Received: date Accepted: date Published: date Publisher’s Note: MDPI stays neu[1]tral with regard to jurisdictional claims in published maps and institu[1]tional affiliations. Copyright: © 2022 by the authors. Submitted for possible open access publication under the terms and conditions of the Creative Commons Attribution (CC BY) license (https://creativecommons.org/license s/by/4.0/). Biomedicines 2022, 10, x  2 of 12 immunological response to vaccines. Among these drugs, steroids are frequently pre- 46 scribed to reduce lung inflammation and to treat wheezing; pwCF also receive frequent 47 antibiotic courses to eradicate gram-negative bacteria from the respiratory airways. Anti- 48 biotics affects gut microbiome and microbiome alteration has been linked to reduced im- 49 munogenicity and efficacy of vaccines [7,8].”

[This is a valuable introduction. Can you provide more specifics from the studies please for the benefit of clinicians and researchers?]

Methods, Results and Conclusions

This is an exemplary straight-forward, well planned, fully reported and well presented study. I have no suggestions for improvement.

Did the software remove the italics from P. aeruginosa?

Exemplary English text

Reviewer 2 Report

I have reviewed the paper by Alicandro et al.

Since besides genetic and cellular factor that may impair the immune response in CF, therapy (such as steroids or antibiotics) may also play a role. Analysis by these in the studied cohort needs to be performed. This was partially done in Table 2 were ICS showed statistically significant difference.

The main limitation of the study is the lack of matched healthy vaccinated controls for comparison. Without this group it is impossible to measure any deviation of what it would be considered a “normal” or expected response to the vaccine.

To save this paper, I would change entirely the directionality of the study, and from the title focus on the effect of ICS in the response to COVID-19 vaccines in pwCF.

Reviewer 3 Report

This study investigated the immune response after two doses of BNT162b2 mRNA-based vaccine among patients with cycstic fibrosis. Overall, the topic is interesting and the manuscript is well written. I just have minor comment.

1. Because more than 2 doses of COVID-19 vaccine was recommended in this ear of Omicron predominance, please add this issue as one of limitaiton.

2.  Please mention the occurence of breakthrough infeciton in this study.

3.  Please add the reference following the sentence "People with CF (pwCF) are considered a clinically vulnerable population and thus 35 they were vaccinated at the very beginning of the vaccination campaign in many countries, including Italy (February-March 2021)"

Round 2

Reviewer 2 Report

Authors did a good job with this new version.